# CAR-T with License to Kill Solid Tumors in Search of a Winning Strategy

**DOI:** 10.3390/ijms20081903

**Published:** 2019-04-17

**Authors:** Benedetto Sacchetti, Andrea Botticelli, Luca Pierelli, Marianna Nuti, Maurizio Alimandi

**Affiliations:** 1Department of Science, University Roma Tre, 00146 Rome, Italy; benedetto.sacchetti@uniroma3.it; 2Department of Clinical and Molecular Medicine, Sapienza University of Rome, 00161 Rome, Italy; andrea.botticelli@uniroma1.it; 3Department of Experimental Medicine, Sapienza University of Rome, 00161 Rome, Italy; luca.pierelli@uniroma1.it (L.P.); marianna.nuti@uniroma1.it (M.N.)

**Keywords:** CAR-T, chimeric antigen receptors, immunotherapy, solid tumors, universal CAR, CD16-CR

## Abstract

Artificial receptors designed for adoptive immune therapies need to absolve dual functions: antigen recognition and abilities to trigger the lytic machinery of reprogrammed effector T lymphocytes. In this way, CAR-T cells deliver their cytotoxic hit to cancer cells expressing targeted tumor antigens, bypassing the limitation of HLA-restricted antigen recognition. Expanding technologies have proposed a wide repertoire of soluble and cellular “immunological weapons” to kill tumor cells; they include monoclonal antibodies recognizing tumor associated antigens on tumor cells and immune cell checkpoint inhibition receptors expressed on tumor specific T cells. Moreover, a wide range of formidable chimeric antigen receptors diversely conceived to sustain quality, strength and duration of signals delivered by engineered T cells have been designed to specifically target tumor cells while minimize off-target toxicities. The latter immunological weapons have shown distinct efficacy and outstanding palmarès in curing leukemia, but limited and durable effects for solid tumors. General experience with checkpoint inhibitors and CAR-T cell immunotherapy has identified a series of variables, weaknesses and strengths, influencing the clinical outcome of the oncologic illness. These aspects will be shortly outlined with the intent of identifying the still “missing strategy” to combat epithelial cancers.

## 1. Introduction

Chimeric Antigen Receptors (CARs) for Adoptive Cell Therapy (ACT) account for specific implementation of functions in a subset of transduced immune effector cells that acquire novel specificities against target cells. In particular, CAR-engineered T lymphocytes are empowered to recognize membrane bound molecules expressed by target cells and trigger a TCR-independent immune reaction against cancer cells, bypassing the Human Leukocyte Antigen (HLA) restriction for antigen presentation.

From the original design where scFv antibodies have been engineered to the T cell receptor (TCR) ζ-chain [1], T-cell redirection strategy has evolved to produce a number of CARs with different signaling abilities that, transduced singularly or in combination, ensure efficient tuning of signals, combinatorial antigen selection and adequate control of toxicity [2]. 

The “state of art” of immunotherapy combines cellular engineering with synthetic biology tools to produce numerous “immune weapons” to be utilized in cancer therapy. The group includes therapeutic monoclonal antibodies (mAbs) directed against Tumor Associated Antigens (TAA), bispecific antibodies, a variety of CARs different for tumor antigen specificity and signaling abilities, and clinical-grade checkpoint inhibitors (ICIs). All these tools are variably utilized to cure different types of liquid and solid tumors, sometimes with remarkable, sometimes with discouraging results.

With the groundbreaking approval of two CAR-T cell therapies, tisagenlecleucel (Kymriah) and axicabtagene ciloleucel (Yescarta) in 2017, the demand for CAR-T cell therapy has increased worldwide with the immediate consequence of dedicating much attention to any aspect of the therapeutic intervention. The effort now is to identify tasks and provide guidelines for Health Care Institutions, Industries and patients to ensure a qualified management of CAR-T adoptive cell therapy towards virtually any kind of tumor. 

For what concerns Research Biology, investigation is now directed to ameliorate CAR-T cell design and manufacturing, with specific aims: (a) to obtain a better control of T cell hyperactivity and exhaustion; (b) to ensure a rapid and flexible intervention for antigen escape; (c) to identify the best targetable tumors.

The first two tasks would be accomplished by studies on CAR engineering. It is evident that structure diversities of CAR intracellular domains (ICDs) impact on signaling abilities and ultimately on T cell functions. CAR ICDs can be designed to deliver signals of different strength, duration and intensity, for the need to amplify or mitigate the immune responses. A direct consequence of CAR-T hyperactivation is the “on target toxicity”, which is mostly related to abundant cytokine release. On the other hand, the “off-target toxicity” is due to the inability of ScFv to distinguish between tumor antigens (expressed on tumor cells) and normal antigens (expressed on normal cells). In any case, excessive spread of signals and uncontrolled reactivity need to be hold in check, and eventually reverted at the appearance of incoming toxicity. 

An opposite, but related problem is T cell exhaustion, which is due to an intrinsic T cell dysfunction. A careful evaluation of scientific reports confirms that, together with antigen escape, T cell exhaustion is a major hurdle faced by patients in trials with CD-19 targeted CAR-T cells. T cell exhaustion is an ipoergic status in which CAR-T cell reactivity falls over time. This is due to decreased transcription of genes associated with memory T cells (IL-6 – STAT3), including antigen stimulation and proliferation, and increased expression of genes involved in T cell effector functions, exhaustion and glucose uptake. 

The other aspect is that conventional CARs have a fixed antigen specificity, a fact that intrinsically harbors the risk for the development of tumor escape variants and limits the efficacy of CAR-T cell therapy due to heterogeneous tumor antigen expression. 

These considerations are now used to improve flexibility of the Chimeric Receptors, redesigning the extracellular domain (ECD) for antigen recognition, and to tune up signaling to better control toxicity and counteract immunosuppression. 

The third task should be taken care by studies on tumor cell biology trying to elucidate the intricate network of dynamics taking place at the immunological synapses, which are regulated by immune checkpoint ligand/receptor interactions. In this context, specificity and complexity of the responses are mediated by tumor cells and its microenvironment, soluble molecules such as IDO1, PD-L1 and IL-10, and cellular components such as tumor-associated macrophages (TAMs) and myeloid-derived suppressor cells (MDSCs), which limit immune cells infiltration and suppress effector T cell activity [3]. It is in fact well known that catabolites released by tumor metabolism have a strong negative impact on T cell function. A number of enzymes like indoleamine-2,3-dioxygenase (IDO), tryptophan-2,3-dioxygenase (TDO), nitric oxide synthase (NOS) and arginase-1 suppress T cells directly (IDO) or indirectly, causing tryptophan and L-arginine T cell deprivation. All these aspects impact also on CAR-T cell functions, as demonstrated in a seminal work describing how increased arginase activity in neuroblastoma impaired NY-ESO-1-specific T-cell receptor and anti-GD2 engineered CAR-T cell proliferation and cytotoxicity [4].

Immune oncology trials on ICIs and CAR-T have identified a number of variables potentially influencing the immunotherapy outcome; among them, some are potentially prognostic, such as: tumor cell biology and mutational burden, presence of an inflammatory tumor microenvironment (TME) recruiting tumor-fighting CD8^+^ T cells, antigen loss and immune escape, fitness of the immune system and timing of administration of the immunotherapy.

Their evaluation will help to identify patients, design guidelines and indicate strategies potentially combining chemo/target therapy, radiotherapy and use of immune checkpoint inhibitors to increase effectiveness of CAR-T cell therapy for solid tumors. 

## 2. Lesson from Immune Checkpoint Inhibitors

### 2.1. Tumor Cell Biology and Mutational Burden 

The latest acquisitions in cancer biology have established the concept that tumor heterogeneity (TH) is highly responsible for unsuccessful anticancer therapies. The reason for that is the ability of many cancer cells to develop even thousands of mutations in their lifetime. Most of these mutations are classified as “passenger mutations” not leading to any biological advantage, while the occurrence of even a low number of “driver mutations”, may dramatically impact on tumor cell progression and be responsible of appearance of new neoplastic phenotypes, over time. 

While errors in the DNA sequence can be possibly repaired by specific repair pathways, cells acquiring aberrant phenotype can be recognized and killed by the immune system reacting against newly expressed tumor-associated antigens (TAAs), called neoantigens [5,6,7,8]. 

For example, in case of genetically unstable colorectal cancers with microsatellite instability (MSI-H) or mismatch-repair deficient (dMMR) tumors, neoplastic foci are often strongly infiltrated by lymphocytes, including activated CD8^+^ T cells [9,10,11], sometime organized in lymphoid tertiary structures [10,11]. This observation has a great impact in clinical practice. Indeed, Dung et al. [12] demonstrated that dMMR cancers are more sensitive to treatment with checkpoint inhibitors, and this led FDA to approve in 2018 Pembrolizumab, as the first drug with tissue/site agnostic indication for patients with MSI-H or dMMR cancers.

Infiltrated T lymphocytes indicative of a potent local immune response have been also observed in pancreatic cancer, colon cancer and in patients with non-Hodgkin lymphoma where high mutational rate generates novel tumor antigens that can be recognized by CD4^+^ T, CD8^+^ T and B cells [13,14,15,16]. In these patients, ICI administration unleashes a clinical appreciable anti-tumor reactivity, where potency depends on a pre-existing population of tumor-primed T cells, targeting “neoantigens” derived by somatic mutations occurring in cancer cells [17,18].

Thus, Tumor Mutational Burden (TMB), initially emerged as an independent biomarker of outcomes across multiple cancer types, was then proposed as selecting criteria in the Checkmate 227 study where Non-Small-Cell Lung Carcinoma (NSCLC) patients with mutational load higher than 10 mutations per megabase demonstrated a benefit of immunotherapy (Nivolumab) [19].

### 2.2. “Cold “and “Hot” Tumors

The tumor microenvironment has important roles in regulating dynamics of cancer/immune cell interactions during tumor progression. The high inflammatory grade of tumor microenvironment is often related to the number of mutations present in malignant cells, which is associated with higher antigenicity; this can be assessed by the presence of tumor-infiltrating CD8^+^ T-cells [5,6] and, sometimes, of histopathological evidences of tumor necrosis. In this case, antigens released from necrotic cells are processed by conventional antigen presenting cells favoring activation of the immune system. An inflammatory/immune microenvironment would probably favor T cell recruitment at tumor side, with increased chances of amplifying the T cell repertoire recognizing tumor antigens. Again, lung cancers occurring in smokers and microsatellite instability in colorectal cancers with higher mutational burden and high-neoantigen load have shown more responsiveness to immune checkpoint inhibitors [18,19,20,21,22]. What is also becoming clear is that primary lesions and metachronous metastases have a heterogeneous immune infiltrate with high mutational diversity, like in metastatic colon cancer, where high Immunoscore correlates with a lower number of metastases [23].

All these observations led to the accepted distinction between “hot” and “cold” tumors. Hot tumors have high mutational burden and a population of infiltrating exhausted PD-1^+^ T cells ready to be expanded in the tumor microenvironment, meanwhile cold tumors are only surrounded by limited number of T cells, mainly localized in the periphery of the tumor, signs of scarce activation and poor T cell penetration [22,23]. 

Careful evaluation of the inflammatory/immune status characterizing hot tumor’s microenvironment would be helpful to identify potentially “good responders” to immunotherapies, including CAR-T cell therapy (Figure 1).

The idea to provoke “ad hoc” a favorable immune environment within the tumor, responds to the necessity of remodeling the microenvironment and its vascular bed to allow better circulation of CAR-T cells at the tumor side, and in the meantime activate the immune system through presentation of newly released antigens by dendritic cells (DC). Radiotherapy, chemotherapy and target therapy inducing immunogenic cell death can potentially revert a “cold tumor” into a “hot tumor” [24,25,26,27,28]. Thus, a sequential strategy, rather than a combination strategy with calculated doses of chemotherapy (metronomic schedule), radiotherapy or target therapy, would be favored in order to prepare patients to receive immunotherapeutic agents or CAR-T adoptive cell therapy [29,30].

## 3. Lesson from Car-T Cell Therapy

### 3.1. Antigen Loss and Immune Escape 

The landscape of mutational signatures designed by next generation sequencing analysis indicates that high mutational burden tumors are the most immunogenic. This is not surprising, and while providing rational for understanding tumor relapse and mechanism of evasion under immunological pressure, sustains the prediction that successful therapies need to quickly counteract the appearance of new neoplastic phenotypes that can occur in any moment during pharmacological treatments. Particularly, immune therapies with TAA–directed mAbs or adoptive cell therapy with engineered CAR-T cells expose tumor cells to a selective pressure favoring the appearance of Ag-negative cells over time.

Antigen loss due to mutations, downregulation or antigen deletion, or even heterogeneous expression of TAAs may favor cancer cell survival, are a prerequisite for tumor relapse [31,32,33,34]. Selection of antigen-negative cells has been described in several clinical studies, including during treatment of B cell malignancies with anti-CD19-CAR or glioblastoma with redirected anti ErbB2-CAR-T cells [35]. 

The need to propose alternative targeting at the appearance of antigen-negative resistant cells have been inspiring the conceptual design of CAR extracellular domains able to bind mAbs or adaptor molecules with TAA associated specificities, to redirect T cell responses against virtually any kind of tumor antigen. In this case, the immunological synapse is established by indirect interaction between CRs and membrane-bound target antigens, through adaptor molecules (AMs) or monoclonal antibodies. Those chimeric receptors are called “Universal CRs” or “Modular CARs (modCARs). 

Among them, the split, universal and programmable (SUPRA) CAR system [36,37], the biotin-binding immune receptor (BBIR CAR) [38,39] and the FcγRIIIa-CR (CD16-CR) represent the most advanced CAR design (Figure 2). They all are able to increase flexibly through different mechanisms, allowing to switch targets without re-engineering the T cells, or multiple targeting, sequentially or in combination, to limit the risks of immune escape for the emergence of antigen-null tumor cells [40,41,42,43,44,45,46]. More importantly, Universal CRs show more capability to regulate potency of T cell activation, for the intrinsic abilities of their own “key-lock” split systems to adjust the affinity for the antigen and for the CR. In this way, signaling strength and activity can be dialed up or down as desired. 

### 3.2. Combining Humoral and Cellular Immune Responses

Clinical successes in cancer treatments based on mAbs demonstrated the relevance of several tumor antigens as therapeutic targets. MAbs for passive cancer immunotherapy directed against overexpressed/hyperactivated signaling receptors or against tumor antigens exert their activity by own mechanism of action, variably interfering with receptor stability, cognate ligand binding or signaling receptor capabilities [47,48]. However, all of them work through the engagement of the antibody-dependent cellular cytotoxicity (ADCC) [49] pathway, and some of them through antibody-dependent cellular phagocytosis (ADCP) [50], complement-dependent cytotoxicity (CDC) [51], antibody-induced apoptosis [52] and antibody-induced programmed cell death [48]. The downstream effect of immune killing is the direct and consequent commitment of the acquired immunity with activation of specific CD4^+^ and CD8^+^ T cells also of the memory lineage [53,54,55].

Trastuzumab (anti-ErbB2 mAb), cetuximab and panitumumab (anti-EGFR mAbs) and rituximab (anti-CD20 mAb) are examples of mAbs of proved efficacy in the treatment of ErbB2^+^ breast cancers, colorectal carcinomas, head and neck cancers and B-cell cell malignancies, respectively [56,57]. On the wave of remarkable successes demonstrated by therapeutic mAbs in cancer treatment the US Food and Drug Administration (FDA) has approved more than 70 antibodies and Fc fusion proteins for the treatment of several diseases, meanwhile more than 570 IgGs, antibodies drug-conjugates and Ab fragments are currently tested in phase I or phase II clinical trials [58]. This robust platform for pre-clinical testing is supported by constant mAbs engineering and novel antigen identification, including neo-antigens, fueling an expanding industrial pipeline for antibody drug discovery.

### 3.3. CD16-CR and Therapeutic mAbs: “Like Two Peas in a Pod”

Solid tumors are not able to promote an efficient ADCC activity for the relative paucity of microenvironment infiltrating NK cells and for the presence of M2 macrophages and immunosuppressive regulatory T cells. Nevertheless, passive administration of mAbs is clinically effective, indicating that strategies aimed to combine therapeutic activities of mAbs with the potential of a T cell-dependent activation at the tumor site might be ideal [41,42,43,44,45,46].

Peculiar to the CD16-CR is the ability to activate engineered T cells only in presence of mAbs opsonized target cells. The overall stability of the immunological synapse (T/cancer cell) that regulates threshold of activation and amplitude of T cell responses depends on two parameters: Chimeric Receptor density and mAb binding affinities. Receptor density regulates thresholds of intracellular T cell activation, whereas the binding affinities of mAbs may influence T cell responses by regulating dynamics of interactions with the tumor antigen and the CD16-CR. Availability of mAbs with different affinities for the antigen and the FcγR binding domain (CD16) will allow for a more precise tuning of engineered T cells activity, to better control possible on- and off-tumor’s reactivities.

On the other end, patient administration of mAbs can be personalized, adjusting frequency and dosage in specific therapeutic windows to maximize cytotoxic killing, or mitigate unwanted side effects. Antibodies can even be withheld to turn off CD16-CR T cell activity for eliminating long-term toxicity. In addition, preclinical studies indicate that CD16-CR T cells can target multiple antigens using a combination of multiple tumor-specific mAbs. These “cocktails” may limit or reduce development of tumor resistance to therapy while increasing sensitivity. It is likely because of these reasons that CD16-CR is the most advanced Universal CR already being tested in clinical trials (NCT03266692-NCT03189836-NCT02776813).

Initial data from Phase I clinical trial evaluating CD16-CR used in combination with rituximab (anti CD20) in adult patients with relapsed/refractory non-Hodgkin lymphoma suggest that CD16-CR T cells can achieve tumor reduction even a low dose, without provoking cytokines release syndrome (CRS) or neurotoxicity.

Therefore, the CD16-CR has the potential to be a powerful tool against highly heterogeneous or genetically unstable cancer types resistant to conventional CAR-T cell therapy, because it reduces the probability of antigen escape. 

## 4. Fitness of the Immune System 

### 4.1. Interplay between Metronomic Chemotherapy and Immunotherapy 

Metronomic chemotherapy, originally designed to overcome drug resistance by shifting the therapeutic target from tumor cells to tumor endothelial cells, is defined as frequent administration of chemotherapeutic agents at a non-toxic dose without extended rest periods. Among the advantages of chronic low-dose drug administration there are the activation of innate and adaptive immunity, induction of tumor dormancy and chemotherapy-driven dependency of cancer cells [59,60]. 

### 4.2. Anti-Angiogenetic Effects 

Neo vascular growth is important to ensure adequate proliferation and metastatic spread of cancer cells, supply of oxygen and nutrients and removal of waste products. On top of that, tumor angiogenesis is characterized by highly permeable and immature vascular structures difficult to mediate T cell adhesion and function. Those are at least few compelling reasons for which tumor angiogenesis is an important target for anti-cancer therapies. Although chemotherapy with conventional drugs targets endothelial cell proliferation and anti-angiogenetic effects are hardly appreciable, simply because vascular endothelial cells quickly recover during treatment breaks. 

Differently from conventional therapy that targets proliferating tumor cell, metronomic therapy is mainly directed to target endothelial cells of the growing tumor vasculature, achieving cancer control by targeting angiogenesis through selective inhibition of endothelial cells migration, proliferation, induction of apoptosis of newly formed tumor microvessels and reduced mobilization of bone marrow-derived endothelial progenitor cells [61,62,63,64,65]. All these effects on the endothelial compartment are extremely valuable, not only for the possibility to preserve its integrity avoiding neo-vascular tumor structures formation, but also for their ability to maximize T cells homing at the tumor site. The other aspect is that low-doses of chemotherapy may preserve immune cells viability, including CAR-T cells, NK cells with ADCC functions and T lymphocytes for CAR-T cell engineering. To this extent, metronomic chemotherapy can exert anti-tumor effects also potentiating host immunity by several immunomodulatory mechanisms [66,67]. 

### 4.3. Immune System

It has been reported that low-dose cyclophosphamide in vitro stimulates macrophages secretion of pro-inflammatory lymphokines, such as IL-6 and IL-12, while down-regulating anti-inflammatory cytokines IL-10 and TGF-β [61]. These effects parallel with a selective reduction of circulating CD4^+^CD25^+^ Tregs and a general depression of their immune suppressive functions through Fox-P3 down-regulation [68,69,70]. Low doses of paclitaxel stimulate myeloid-derived suppressor cells (MDSCs) to differentiate into functionally immune stimulatory DCs in an independent Toll-like receptor (TLR)4 activation [71]. Other effects on DCs maturation, including a more efficient antigen-presenting activity, are due to increased expression of CD83, CD80 and CD40. On the other end, paclitaxel and doxorubicin administered at non-cytotoxic doses activate the MHC class I antigen processing machinery in colon cancer cells, making them more susceptible to T lymphocytes cytotoxic hits [72]. 

The concepts expressed above support the adoption of metronomic chemotherapy as ideal to conjugate with immune therapy, including CAR-T adoptive cell therapy.

### 4.4. The Right Therapy, for the Right Patient, at the Right Moment

In the time of precision medicine, new postulates can be successfully adopted for immunotherapy and adoptive cell therapy. Several studies have shown that gut microbiome may influence anti-tumor immune responses via innate and adaptive immunity, and that therapeutic responses may be improved via its modulation [73,74] Diversity on bacterial composition has a general impact on the immune system “fitness”, as indicated by increased antigen presentation and improved effector T cell functions in patients with “favorable” gut microbiome (abundant of Ruminococcaceae/Faecalibacterium) as compared to “unfavorable” gut microbiome (highly abundant of Bacteroidales). These findings have now been consolidated and impact on progression free survival and toxicities rate in patients affected by solid tumors treated with anti PD-1 and anti CTLA-4 [73,75].

Positive effects have been demonstrated in cancer patients with “favorable” gut microbiome, where enhanced anti-tumor immune responses in the periphery and within the tumor microenvironment are sustained by increased lymphoid and myeloid intra-tumor infiltration and superior antigen presentation [73,76,77,78,79].

While mechanisms are still unclear, the microbiota is thought to differently interact with specific immune components to induce conversion from non-inflammatory to pro-inflammatory environments through synthesis of regulatory cytokines and, in some cases, influencing the PD-1-PD-L1 signaling axis, holding in check the activation/inhibition balance of the immune responses [74,75,76,79,80,81]. Those aspects should not be neglected, since intra-tumor infiltration, antigen presentation, cytokines secretion may help tumor infiltrating lymphocytes and CAR-T resilience from the inhibitory effects imposed by the tumor microenvironment. 

### 4.5. Timing of Administration of Immunotherapy

Another question is: which is the best timing to infuse patients with CAR-T cells to get maximal clinical effects while minimizing toxicity or adverse effects? Experience with checkpoint inhibitors therapy indicates that immunological changes could be detected quite early in the peripheral blood of patients, suggesting that, favorable impact on immunological repertoire, control of toxicity and effects on tumor shrinking may benefit of an early biomarker assessment after ICI administration. Ultimately, early intervene allows for better monitoring of the immune fitness, timely decision for correct direction to take and better understanding of the immune treatment outcome. Another point to consider is tumor burden. It is quite accepted among investigators that there is a strong link between the degree of immune expansion of T cells/tumor burden/clinical outcome. The rationale for this is easily identifiable in the relatively low number of tumor cells composing tumor burden at its early stage that can be reached “in loco” by immune cells.

## 5. Discussion

Until a recent past, the focus of cancer treatment was addressed in the assumption of killing tumor cells using maximal therapeutic doses (MTD) of chemotherapy and radiotherapy, with results sometime disheartening, sometime encouraging, but with sequela of toxicities and of a general immunocompromised status.

The introduction of ICIs in clinical and the brilliant results from CAR-T cell therapy in hematologic tumors have revolutionized the entire scenario of the immune-oncology field, up to indicate in the immunotherapy a potent inalienable tool for cancer treatment.

This view reverts the previous paradigm centered on the killing of tumor cells directly, in favor of a more sophisticated treatment addressed to utilize the intrinsic abilities of the immune system, particularly T cells, to recognize and kill cancer cells. It is evident that immunotherapy has better anti-tumor results when used in combination with other therapies that have immunomodulation abilities. The ideal treatment has to calibrate desired and unwanted effects, managing between killing the bad cells (cancer cells) and sustaining the good cells (immune and endothelial cells).

Recent systematic meta-analyses conducted up to June 2018 to investigate the efficacy of CAR-T cell therapy in solid tumors indicate that the pooled response rate of CAR-T cell therapy in solid tumor reaches 9%, with the best therapeutic effects observed in neuroblastoma (response rate: 33%) [82,83].

Although these encouraging results, CAR-T cell engineering for solid tumors need further development, overall in the direction to facilitate recruitment, protection and activation of the immune cells potentially able to eliminate their target. On the other hand, it is necessary to elaborate synergistic multi-task therapies aimed to reduce tumor burden; avoid tumor angiogenesis; increase the release of neo-antigens to stimulate immune responses; maintain a healthy immune cell compartment for optimal CAR-T cell responses, while decreasing number and activity of immunosuppressive cells.

A satisfactory activation of the immune system depends on multiple parameters. Some of them are generally depending on age, obesity and sex hormones that, through induction of a proinflammatory state, sustain growth and expansion of suppressive immature myeloid cells. More important are the general fitness of the immune system and the biology of the tumor; together they determine by multiple means the composition of the immune infiltrate at the tumor site and the characteristics of the inflammatory/immune response in its microenvironment.

Would it be possible the conversion of a “cold” tumor into a “hot” tumor? This has already been demonstrated in patients with glioblastoma. Analysis of TCR clonality pre and post anti-EGFRvIII CAR-T cell infusion revealed intense recruitment of endogenous non-modified T cells through chemokines production, indicating a good activation of the immune system in response to increased neo-antigens production [84]. This, although with consciousness, provides the rational to combine ICIs to adoptive CAR-T cell therapy [85,86,87].

The Design of CARs has unlimited potentials, and enough observation can potentially inspire future design of chimeric receptors, both for EC and IC domains. ICDs can be designed to sustain a less-differentiated T cell phenotype, with enhanced capacity and diminished propensities for exhaustion. Transcriptomic profiling from chronic lymphocytic leukemia patients successfully treated with CAR-T cells, revealed a general implementation of the IL-6/STAT3 signaling pathways, as indicated by increased production of IL-6, IL-17, IL-22, IL-31 and CCL20 [88,89]. This provides a rational for engineering new ICDs able to sustain and maximize T cell functions, promoting the expansion of clinically potent memory T cells.

Antigen loss, or simply its downregulation, has inspired the development of CARs targeting more than one antigen. Among “Universal CARs”, the CD16-CR appears to offer the most reliable approach to treat solid tumors with CAR-T, for the ability to combine T cell redirection with the therapeutic activity of mAbs, for the possibility to quickly shift target antigen at the appearance of a new neoplastic phenotype, and for the possibility to utilize IgG possibly produced by B cells against tumor antigens.

The other aspect to be considered is how to maximize the activity of CAR-T cells in a hostile suppressive TME [90]. Encouraging results come from IDO inhibitors (fludarabine and cyclophosphamide) which demonstrated to enhance CAR T-cell efficacy against CD19 B-cell leukemia and are now being evaluated in clinical trials [91,92,93]. If validated, this preliminary information will provide a good rationale for combining the two immunotherapies.

CAR-T cells have been engineering to secrete and deliver PD-1, CTLA-4 or PD-L1 antibodies at the tumor site [94,95,96]. This strategy has shown promising results in mice models, and has been adopted in clinical trials for MUC1, EGFR, EGFRvIII and mesothelin expressing tumors [97,98]. This combination has the potential to counteract the exhaustion of the CD8^+^ TILs, CAR-T cells and other immune cells accumulating at the tumor site.

The last consideration is for Universal CARs and ICIs. Although still speculative, a further possibility to use ICIs with CAR-T cell therapy, at least in selected cancer types, is offered by the possibility to redirect CD16-CR engineered T-cells to neoplastic cells by PD-L1 IC-blocking mAbs. This would have the advantage of maximizing T-cell reactivity by blocking the SHP2-mediated dephosphorylation of proximal TCR signal transducers, while triggering ADCC-like activity in CD16-CR transduced T-cells against PD-L1^+^ cancer cells, but no experiments have been done in this direction.

In perspective, how does one prepare patients to receive immunotherapy: a combination of therapies or in sequence? It would depend on drug-specific immunological effects, according to the specific tumor type and phase of cancer-immune cycle in which they act. A prototype diagram is proposed in Figure 3.

To conclude, the body of information cumulated from recently finished and ongoing immuno-oncology clinical trials have produced significant advance in personalization of immunotherapy. Despite this, only few trials have given impactful insights in clinical practice and much has to be done to maximize patient benefits. CAR-T adoptive cell therapy for solid tumors is only at the onset of a promising scenario, which is dominated by the perspective of a cross-disciplinary approach made of a multitask combination of therapies.

## Figures and Tables

**Figure 1 ijms-20-01903-f001:**
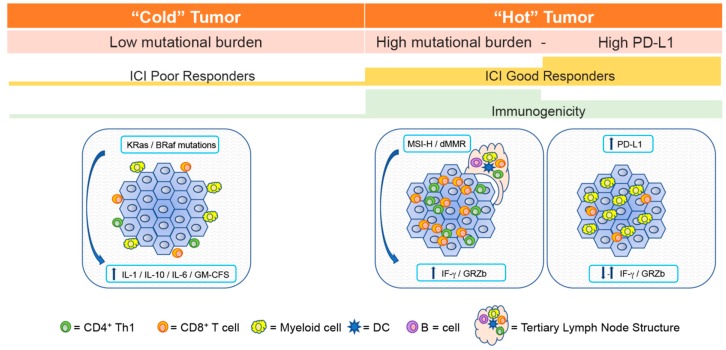
and “Hot” Tumors. Tumor microenvironments can be classified as immunologically “hot” (high immunogenicity) or “cold” (low immunogenicity), depending on tumor antigenicity, mutational burden, PD-1 and PD-L1 expression and lymphocyte infiltration. Histological evaluation of these parameters allows prediction of ICI responses. “Cold” tumors (I-E: Infiltrated Excluded) are characterized by poor CTL infiltration and by the presence along the tumor margins of tumor-associated macrophages (TAMs), likely preventing CTL infiltration into the core; these tumors are characterized by a frequent hyper-activation of the MAPK (Mitogen-Activated Protein Kinase) pathway, often maintained by Kras and Braf mutations, that sustains transcription of immunosuppressive lymphokines. “Hot” tumors (I-I: Infiltrated-Inflamed) are either characterized by PD-1 ligand (PD-L1) expression and high infiltration of PD-1 expressing CTLs, or by high mutational burden sustaining neo-antigens production and high infiltration of leukocytes, sometime organized in tertiary lymph node structures.

**Figure 2 ijms-20-01903-f002:**
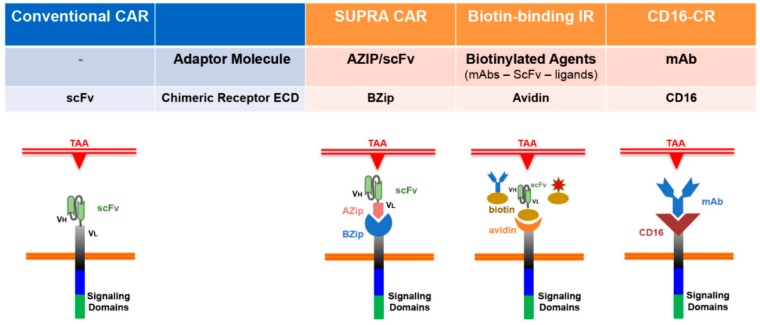
Structural design of Universal Chimeric Receptors: comparison versus canonical CARs. SUPRA CAR system is a universal receptor consisting of two-components: a leucine zipper adaptor (zipCAR) on T cells and a separate scFv with leucine zipper adaptor (zipFv) molecule targeting specific antigens. The biotin-banding immune receptor contains dimeric avidin (dcAv BBIR) able to bind a variety of biotinylated antigen-specific molecules (scFV, mAbs or tumor-specific ligands) expressed on the surface of target cells. The BBIR “lock-key” mechanism is based on binding of biotin to avidin. The CD16 binding module of the CD16-CR is combined the transmembrane and signaling domains of the CR. Virtually any mAb can redirect FcγR-CR T cells toward TAAs expressed by malignant cells.

**Figure 3 ijms-20-01903-f003:**
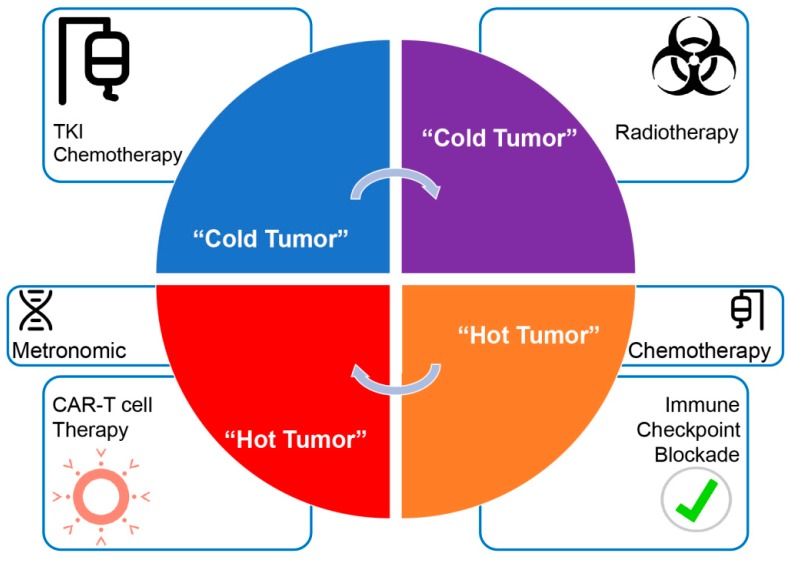
The focus of treatment shifts from killing tumor cells to treating the specific biologic characteristics of the tumor and its environment; this increases the intrinsic abilities of the immune system to combat cancer. Timing of ICIs and CAR-T cells therapy administration has to coincide with a high inflammatory tumor microenvironment to ensure presence and effectiveness of ICI-induced potential tumor-fighting CD8^+^ T lymphocytes and redirected CAR-T cells. Increased DNA damage and tumor necrosis provoked by prior conventional chemotherapy and radiotherapy may increase neo-antigen release and DCs processing leading to increased tumor antigenicity; tyrosine kinase inhibitors (TKI) administration may contribute to reduce the transcriptionally controlled secretion of immunosuppressive cytokines in the TME, while participating to cancer cell death; adopted metronomic chemotherapy may help survival and functionality of immune cells while maintaining efficiency of the endothelial-cells/vessels compartment during immunotherapy.

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
