# Peer review of "CAR-T with License to Kill Solid Tumors in Search of a Winning Strategy"

_ijms, 2019, doi:10.3390/ijms20081903_

Round 1
Reviewer 1 Report
This review offers a comprehensive state of the -art of the CAR biology, trying to place it in the more broad context of immunotherapy. Nothing really new, but it s well written overall and quite up-to date.
Few missing papers from the reference list that are important and add value to this review:
Galon J, Bruni D. Approaches to treat immune hot, altered and cold tumours with combination immunotherapies. Nat Rev Drug Discov. 2019 Mar;18(3):197-218. doi: 10.1038/s41573-018-0007-y. Review. PubMed PMID: 30610226.
Van den Eynde M, Mlecnik B, Bindea G, Fredriksen T, Church SE, Lafontaine L, Haicheur N, Marliot F, Angelova M, Vasaturo A, Bruni D, Jouret-Mourin A, Baldin P, Huyghe N, Haustermans K, Debucquoy A, Van Cutsem E, Gigot JF, Hubert C, Kartheuser A, Remue C, Léonard D, Valge-Archer V, Pagès F, Machiels JP, Galon J. The Link between the Multiverse of Immune Microenvironments in Metastases and the Survival of Colorectal Cancer Patients. Cancer Cell. 2018 Dec 10;34(6):1012-1026.e3. doi: 10.1016/j.ccell.2018.11.003. PubMed PMID: 30537506.
Angelova M, Mlecnik B, Vasaturo A, Bindea G, Fredriksen T, Lafontaine L, Buttard B, Morgand E, Bruni D, Jouret-Mourin A, Hubert C, Kartheuser A, Humblet Y, Ceccarelli M, Syed N, Marincola FM, Bedognetti D, Van den Eynde M, Galon J. Evolution of Metastases in Space and Time under Immune Selection. Cell. 2018 Oct 18;175(3):751-765.e16. doi: 10.1016/j.cell.2018.09.018. Epub 2018 Oct 11. PubMed PMID: 30318143.
Author Response
We really thank the reviewer for the appreciation of our work and for the helpful comments meant to improve the quality of the manuscript.
As suggested by the reviewer we add five fundamental papers needed to be mentioned in the text and bibliography, to better describe the landscape of CAR-T cell therapy in solid tumors.
Galon, J.; Bruni, D.; Approaches to treat immune hot, altered and cold tumours with combination immunotherapies. Nat Rev Drug Discov. 2019; 18(3):197-218.
Van den Eynde, M.; Mlecnik, B.; Bindea, G.; Fredriksen, T.; Church, S.E.; Lafontaine, L.; Haicheur, N.; Marliot, F.; Angelova, M.; Vasaturo, A.; et al. The Link between the Multiverse of Immune Microenvironments in Metastases and the Survival of Colorectal Cancer Patients. Cancer Cell.2018; 34(6):1012-1026.e3.
Angelova, M.; Mlecnik, B.; Vasaturo, A.; Bindea, G.; Fredriksen, T.; Lafontaine, L.; Buttard, B.; Morgand, E.; Bruni, D.; Jouret-Mourin, A.; et al. Evolution of Metastases in Space and Time under Immune Selection. Cell. 2018; 175(3):751-765.e16.
Martinez, M.; Moon, E.K. CAR T Cells for Solid Tumors: New Strategies for Finding, Infiltrating, and Surviving in the Tumor Microenvironment.Front Immunol. 2019; 10:128. doi: 10.3389/fimmu.2019.00128.
Heyman, B.; Yang, Y. Chimeric Antigen Receptor T Cell Therapy for Solid Tumors: Current Status, Obstacles and Future Strategies.Cancers (Basel). 2019; 11(2). pii: E191. doi: 10.3390/cancers11020191.
Minutolo, N.G.; Hollander, E.E.; Powell Jr., D.J.The Emergence of Universal Immune Receptor T Cell Therapy for Cancer. Front. Oncol.2019; https://doi.org/10.3389/fonc.2019.00176
Reviewer 2 Report
The use of chimeric antigen receptor (CAR) T cells is gaining traction as one of the most promising advances in cancer immuno-therapy. In this review authors talk about the CAR T cells for solid tumors. They touched upon very interesting and burning topic. Their literature search related to CAR T is very informative and well presented, but not sufficient. Authors missed to mention many of the newly published review in this manuscript (Marina Martinez 2019, Heyman 2019). So please go through latest review cite their information too. Please present so far status of clinical trial using CAR T cells in table form. Over all this manuscript can be accepted after minor revision as I suggested.
Author Response
We really thank the reviewer for the appreciation of our work and for the helpful comments meant to improve the quality of the manuscript.
As suggested we added five fundamental papers needed to be mentioned in the text and bibliography, to better describe the landscape of CAR-T cell therapy in solid tumors.
In regard of providing a table presenting clinical trials using CAR-T cells as suggested by reviewer two, we have been tempted to do it from the beginning, at the first round of submission. We didn’t do that, for several reasons:
1) The structure of our manuscript is more a commentary, a perspective, a current opinion than a real review. Our intent is to offer to the reader new elements in a comprehensive scenario possibly leading to a different way of thinking CAR-T cell therapy for solid tumors.
2) In our manuscript we never refer to past experience of CAR-T cell trials in solid tumors, and it will be difficult to include a table, that will be necessary only if we include a new chapter or section in the manuscript in support of the table.
3) If accepted our manuscript will be part of a Special Issue "CAR-T Cell Therapy" with the contribution of several authors in the field. We reasoned that a section with past and current experience in clinical trials using CAR-T cells with the inclusion of a supportive table, would be possibly included in other manuscripts, part of this Special Issue.
For these reasons, but with the permission of the reviewer, we ask for a second thought, allowing us to leave intact the spirit of our manuscript.